# Role of framework mutations and antibody flexibility in the evolution of broadly neutralizing antibodies

Victor Ovchinnikov[1†], Joy E Louveau[2†], John P Barton[3,4,5,6†‡], Martin Karplus[1,7]*, Arup K Chakraborty[3,4,5,6,8,9]*

[1]Department of Chemistry and Chemical Biology, Harvard University, Cambridge, United States; [2]Harvard-MIT Division of Health Sciences and Technology, Massachusetts Institute of Technology, Cambridge, United States; [3]Department of Chemical Engineering, Massachusetts Institute of Technology, Cambridge, United States; [4]Department of Physics, Massachusetts Institute of Technology, Cambridge, United States; [5]Institute for Medical Engineering and Science, Massachusetts Institute of Technology, Cambridge, United States; [6]Ragon Institute of MGH, MIT and Harvard, Cambridge, United States; [7]Laboratoire de Chimie Biophysique, ISIS, Universite de Strasbourg, Strasbourg, France; [8]Department of Biological Engineering, Massachusetts Institute of Technology, Cambridge, United States; [9]Department of Chemistry, Massachusetts Institute of Technology, Cambridge, United States

*For correspondence:
marci@tammy.harvard.edu (MK);
arupc@MIT.EDU (AKC)

[†]These authors contributed equally to this work

Present address: [‡]Department of Physics and Astronomy, University of California, Riverside, United States

**Abstract** Eliciting antibodies that are cross reactive with surface proteins of diverse strains of highly mutable pathogens (e.g., HIV, influenza) could be key for developing effective universal vaccines. Mutations in the framework regions of such broadly neutralizing antibodies (bnAbs) have been reported to play a role in determining their properties. We used molecular dynamics simulations and models of affinity maturation to study specific bnAbs against HIV. Our results suggest that there are different classes of evolutionary lineages for the bnAbs. If germline B cells that initiate affinity maturation have high affinity for the conserved residues of the targeted epitope, framework mutations increase antibody rigidity as affinity maturation progresses to evolve bnAbs. If the germline B cells exhibit weak/moderate affinity for conserved residues, an initial increase in flexibility via framework mutations may be required for the evolution of bnAbs. Subsequent mutations that increase rigidity result in highly potent bnAbs. Implications of our results for immunogen design are discussed.
DOI: https://doi.org/10.7554/eLife.33038.001

## Introduction

The HIV/AIDS epidemic affects more than 37 million individuals worldwide, and there were 2 million new infections in 2014 (*Deeks et al., 2015*). The introduction of antiretroviral drugs in the 1990s has made HIV infection a manageable condition, but less than half of the persons diagnosed with HIV remain in care in the United States. In sub-Saharan Africa, the epicenter of the disease, this problem is far more acute. Vaccination is a way to confront this challenge and eradicate the disease. One major obstacle to developing a universal vaccine is the high mutability of HIV (*Korber et al., 2001*). In order to prevent new infections, the vaccine-induced immune response must protect against a great diversity of circulating HIV strains. Antibodies (Abs) that can neutralize a broad diversity of HIV strains, known as broadly neutralizing antibodies (bnAbs), are of great interest in this regard because

they have shown potential to both prevent new infection in animal models and to control existing infection for some duration in both human and animal models (*Barouch et al., 2013*; *Klein et al., 2012*; *Mascola et al., 2000*; *Moldt et al., 2012*; *Lu et al., 2016*).

BnAbs naturally evolve in only a subset of HIV-infected patients, and usually in low titers after several years of infection. The isolation of bnAbs from patients provides proof that the human immune system can evolve such antibodies. Past attempts at eliciting them by vaccination have failed (*Kong and Sattentau, 2012*; *McCoy and Weiss, 2013*). Many efforts are currently underway to design immunogens and vaccination strategies that can induce bnAbs. Despite significant progress (*Shaffer et al., 2016*; *Wang et al., 2015*; *Escolano et al., 2016*; *Steichen et al., 2016*; *Bonsignori et al., 2016*), a vaccination protocol that can efficiently induce bnAbs in non-human primates or humans is not available.

The key process in the development of Abs is a Darwinian evolutionary process known as affinity maturation (*Shlomchik and Weisel, 2012*; *Victora and Nussenzweig, 2012*). A B cell is activated upon binding of its B cell receptor (BCR) to a part of the proteins (epitope) that constitutes the antigen (e.g., spikes on the surface of viruses). Activated B cells can seed structures called germinal centers (GCs) in lymph nodes where affinity maturation occurs. During affinity maturation (AM) B cells proliferate, and upon induction of the activation-induced cytidine deaminase (AID) gene, mutations are introduced in the receptor at a high rate (somatic hypermutation). The B cells with mutated BCRs then interact with the antigen displayed on the surface of Follicular Dendritic Cells (FDCs). The B cells undergo selection to favor those with receptors that bind more strongly to the target epitope. Upon immunization with a single antigen, as cycles of diversification and selection ensue in the GC, antibodies with increasingly higher affinity for the antigen are produced (*Eisen and Siskind, 1964*).

For an HIV vaccine to produce bnAbs, immunization with a single strain of antigen will likely not suffice as it would lead AM to produce strain-specific antibodies. Immunization with multiple variant antigens that have the same amino acid sequence at conserved positions on the viral spike proteins, but diverse amino acids at the surrounding positions, are likely to be required. An example set of conserved residues is the CD4 binding site, which is a part of the epitopes targeted by certain bnAbs. Recent studies have suggested that the variant antigens can act as conflicting selection forces, which frustrate the Darwinian evolutionary process of AM under some conditions (*Shaffer et al., 2016*; *Wang et al., 2015*). Under some circumstances, the importance of sequential immunization with variant antigens for the induction of bnAbs has been shown (*Wang et al., 2015*; *Escolano et al., 2016*; *Steichen et al., 2016*), and the possibility of designing optimal cocktails of variant antigens has also been noted (*Dosenovic et al., 2015*; *Shaffer et al., 2016*).

BnAbs exhibit unusual structural features such as a high degree of somatic hypermutation (including insertions and deletions) in the antigen binding regions known as the complementarity determining regions (CDRs), as well as changes in the surrounding framework regions (FWR) (*Scheid et al., 2009*; *Sok et al., 2013*). Most efforts to understand the evolution of bnAbs have concentrated on mutations in the CDRs. However, mutations in FWRs that are spatially separated from the antigen-binding site may also affect binding properties. For example, Klein et al. showed that, for the bnAbs they studied, the reversal of FWR mutations reduces breadth and potency. However, the FWR mutations had no effect on non-broadly neutralizing antibodies (*Klein et al., 2013*). This finding implies that FWR mutations can be important for the broadly neutralizing activity, and it has been proposed that FWR mutations increase neutralizing breadth by providing the Ab with greater conformational flexibility (*Scheid et al., 2011*; *Klein et al., 2013*). A subsequent study of the 4E10 bnAb also found a high degree of plasticity in the mature antibody (*Finton et al., 2014*).

The implications of these studies contradict the established paradigm of antibody maturation, which suggests that antibodies progressively become more rigid as AM ensues (*Schmidt et al., 2013*; *Eisen and Chakraborty, 2010*; *Foote and Milstein, 1994*; *Wedemayer et al., 1997*; *Thorpe and Brooks, 2007*). Since bnAbs evolve upon undergoing AM induced by multiple variant antigens, different selection forces may be at play.

A recent study on a class of enzymes showed that adaptation to different ligands in a PDZ domain preferentially occurred not only through mutations in the ligand-binding site, but rather through a collection of distant mutations (*Raman et al., 2016*). These mutations worked allosterically to enable multiple conformational states, therefore allowing binding to a diversity of ligands during adaptation. This finding, and those reported by Nussenzweig and co-workers (*Scheid et al., 2011*;

*Klein et al., 2013*), led us to hypothesize that mutations in the framework region (FWR) of BCRs could potentially affect the conformational states of the antigen binding region and allow BCRs to bind to a broad range of variant antigens present during AM induced by natural infection or vaccination. This may, in turn, enable B cells to better negotiate the conflicting selection forces represented by the variant antigens and thus facilitate the evolution of bnAbs.

To explore this hypothesis, we used molecular dynamics (MD) simulations to quantify the structural flexibility of antibodies obtained at different stages of AM in three bnAb lineages. These studies were augmented by a simplified computational model of AM in the presence of multiple variant antigens that simulated the effect of mutations in both the CDR and the FWR to provide insight on the evolution of flexibility and its influence on breadth.

Our results suggest that distinct evolutionary pathways are followed during the evolution of breadth and potency in bnAbs, and we provide mechanistic insights into the underlying reasons. If the binding affinity of the germline BCR to the conserved residues shared by the variant antigens is high, the traditional paradigm where the antibodies become progressively more rigid as AM progresses is predicted. If this is not the case, FWR mutations that increase flexibility are favored. Additional complexities are predicted as breadth and potency evolve. From the standpoint of designing vaccination protocols, our results imply that, if model antigens can prime germline B cells whose BCRs bind strongly to the shared conserved residues of the boosting variant antigens, inducing FWR mutations that influence receptor flexibility is not essential for the evolution of bnAbs. This is significant as it simplifies the task of immunogen design.

## Results

### MD simulations of different bnAb lineages show varying effects of framework mutations

#### Description of the MD simulations

We used atomistically detailed molecular dynamics (MD) simulations to study how flexibility evolved in three different bnAb lineages (see *Table 1*): bnAbs 3BNC60 and CH103 bind to the CD4 binding site on the HIV gp120 protein, and bnAb PGT121 binds primarily to glycans near the V3 chain of gp120. We chose these specific lineages because of the availability of high-resolution crystal structures of germline, intermediate, and mature antibodies.

**Table 1.** Structures of the broadly neutralizing antibodies investigated in this study.

For each lineage, three structures are considered: the matured Ab, an intermediate maturity Ab, and an inferred unmutated germline structure (the exact sequence of the germline B cell that seeds a GC is difficult to know as some junctional diversity could be introduced prior to the GC reaction). Resolution of the structures is indicated after the PDB ID. Only the variable chains of the antibodies were simulated (see Materials and methods).

| Lineage | Maturity | Pdb id (Å) |
|---|---|---|
| 3BNC60 | Matured | 3RPI (*Scheid et al., 2011*) (2.64) |
| | Germline | 5F7E (*Scharf et al., 2016*) (1.90) |
| | P61A reversal* | 4GW4 (*Klein et al., 2013*) (2.65) |
| CH103 | Matured | 4JAN (*Fera et al., 2014*) (3.15) |
| | UCA | 4QHK (*Fera et al., 2014*) (3.49) |
| | Chimera† | 4QHL (*Fera et al., 2014*) (3.15) |
| PGT121 | Matured | 4FQ1 (*Mouquet et al., 2012*) (3.0) |
| | Germline | 4FQQ (*Mouquet et al., 2012*) (2.4) |
| | Chimera‡ | 5CEZ (*Garces et al., 2015*) (3.0) |

*Single mutation reverted from bnAb;
†Intermediate HC with the LC from the unmutated common ancestor (UCA);
‡Intermediate HC 3 hr with mature LC 109L.
DOI: https://doi.org/10.7554/eLife.33038.002

We generated five 100ns trajectories for each structure, starting from different initial conditions (velocities; see Materials and methods). The root mean square distances (RMSD) from the initial structures are shown in *Figure 1—figure supplement 1*. The RMSD time series for each of the five trajectories are concatenated to show that trajectory differences are significant, thus indicating that the trajectories are not correlated and each is meaningful. With the exception of the heavy chain (HC) of the mature 3BNC60, the RMSD are under about 1.5 Å, indicating that the structures are stable in the simulation. Examination of the 3BNC60 trajectory showed local $\beta$-sheet instability near residue P61 which leads to partial unfolding of a beta hairpin in simulation 5 (see *Figure 1—figure supplement 1A*, and also *Figures 1A* and *2A*, discussed below). This result is consistent (see below) with experiments (*Klein et al., 2013*). The RMSD graphs also provide a qualitative measure of antibody flexibility. For example, in the 3BNC60 lineage, the HC shows progressively higher RMSD going from the germline (GL) to the mature structure. The reverse is observed for the CH103 HC, although the differences appear smaller than in the 3BNC60 case. The PGT121 lineage exhibits more complexity that is described below in quantitative terms.

To assess the flexibility quantitatively as a function of residue number, the structures obtained from the all-atom simulations were coarse-grained to a representation of one bead per residue (see Materials and methods) and RMS positional fluctuations from the mean bead position were computed for each bead. More flexible molecules are characterized by larger RMS fluctuations. Further, the standard deviation in the RMSF plots allows one to distinguish between antibody regions that are flexible but stably folded (small Std. Dev.) and antibody regions that are either partially unfolded or have significant conformational heterogeneity across the simulations (high Std. Dev). This coarse-

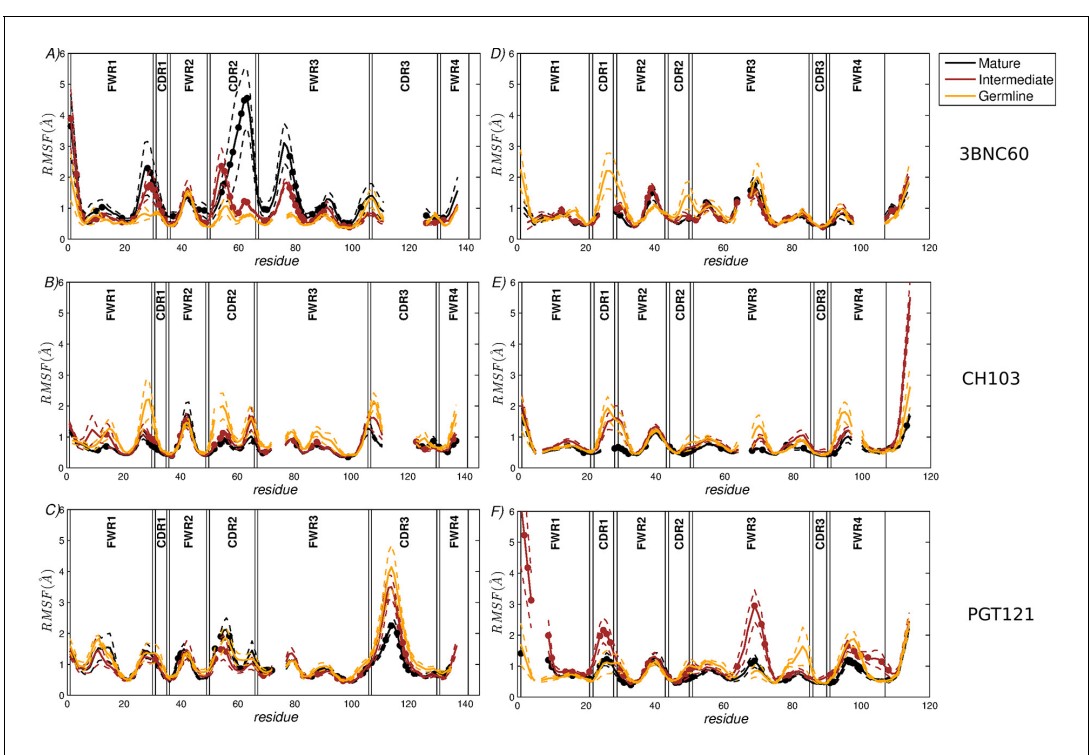

**Figure 1.** RMSF of the CG residue model of the heavy chains (A–C) and the light chains (D–F). (A,D) 3BNC60 lineage; (B,E) CH103 lineage; (C,F) PGT121 lineage. In order to compare the three lineages, all sequences are multiply aligned. This procedure creates gaps in the traces corresponding to antibodies with shorter loops in the region; i.e., there are no actual missing residue coordinates. The dashed lines bound the region of one standard deviation above and below the average trace. Bullets indicate mutations acquired during AM. The definitions of FWR and CDR regions are taken from (*Scheid et al., 2011*).

DOI: https://doi.org/10.7554/eLife.33038.003

The following figure supplement is available for figure 1:

**Figure supplement 1.** Root mean square distance between simulation and initial structures for the 3BNC60 lineage.

DOI: https://doi.org/10.7554/eLife.33038.004

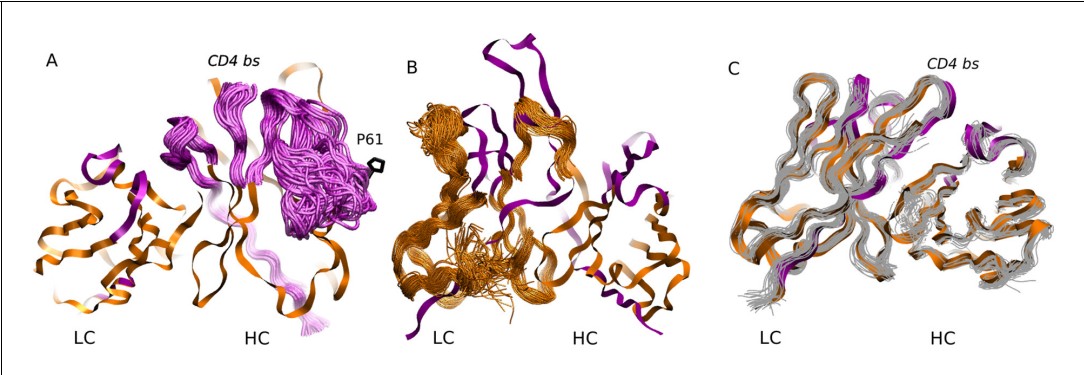

**Figure 2.** MD simulation structures of antibodies. (**A**) Simulation structure of the mature 3BNC60 antibody. Antibody framework regions are shown in orange, and the CDR regions are in purple. The conformational flexibility of the CDR regions of the heavy chain is illustrated by overlaying 24 conformations of the CDRs in 4ns intervals from each of the five trajectories. The P61 residue is shown in black; (**B**) simulation structure of the intermediate PGT121 antibody. The structure is colored as in (**A**), except that the conformational flexibility is illustrated for the FWR region of the light chain using 24 overlaid orange curves; (**C**) simulation structure of the mature CH103 antibody. Twenty-four gray curves are overlaid to illustrate the relatively lower overall flexibility of this antibody.

DOI: https://doi.org/10.7554/eLife.33038.005

The following figure supplements are available for figure 2:

**Figure supplement 1.** Interactions between the CDR1 and FWR3 regions of the PGT121 light chain.

DOI: https://doi.org/10.7554/eLife.33038.006

**Figure supplement 2.** Structure of antibody 3BNC60 (*Scheid et al., 2011*).

DOI: https://doi.org/10.7554/eLife.33038.007

graining also allowed us to calculate the conformational entropy of the antibodies, which is a quantitative thermodynamic measure of their overall flexibility. Toward this end, we used standard quasi-harmonic analysis which treats each bond as a spring (see Materials and methods). Although it is known that the conformational entropy from quasi-harmonic analysis typically overestimates the true conformational entropy (see, e.g., [*Tyka et al., 2007*]), the relative differences between the structures studied are expected to be meaningful because of the overall structure similarity of antibodies in the same lineage. More flexible molecules are characterized by a larger value of the conformational entropy.

## Results of the MD simulations

The results shown in *Figure 1A* quantify the dramatic effect of the P61A reversal mutation in the heavy chain of 3BNC60. Consistent with experiments, the presence of proline in the mature antibody disrupts the secondary structure of the $\beta$-sheet connecting the CDR2 and FWR3 regions, resulting in partial unfolding of the region (*Klein et al., 2013*). The magnitude of the fluctuations in the CDR regions is illustrated qualitatively in *Figure 2A* by overlaying multiple CDR conformations. The fluctuations are highest in the region surrounding the P61 residue. The high fluctuations propagate to other regions (e.g. FWR1/CDR1 and FWR3) because of geometric (but not sequence) proximity (*Figure 1*). The large RMS fluctuations indicate a more flexible structure, and are thus consistent with the observation that the melting temperature for the P61A reverted 3BNC60 mature antibody was 5K lower than that for the mature bnAb (*Klein et al., 2013*). Although the P61A single reversal has a dramatic effect on the dynamics of the Ab, it is clear from *Figure 1A* that other somatic mutations also increase the flexibility of the 3BNC60 heavy chain relative to the germline structure. This is made clear from the values of the calculated conformational entropy (*Figure 3*), where we observe a progressive increase in the conformational entropy from the germline to intermediate to mature antibody for the 3BNC60 lineage. The conformational entropy of the light chains of the 3BNC60 lineage of antibodies show a modest decline (*Figure 3*). From these results we conclude that there was a progressive increase in flexibility of the 3BNC60 lineage of antibodies as affinity maturation progressed.

The evolution of flexibility of the heavy chains of the CH103 lineage is opposite to that of 3BNC60 (*Figure 1B*), since the antibodies become progressively stiffer as the lineage matures. This

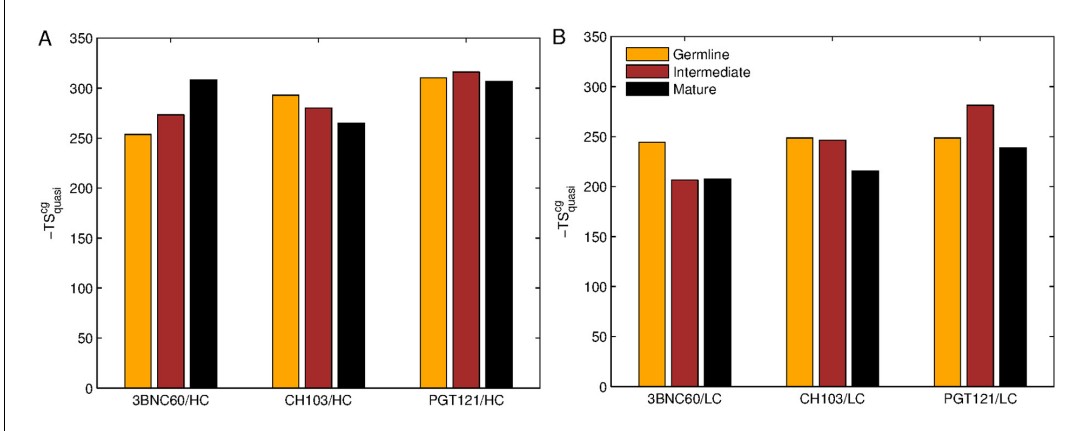

**Figure 3.** Evolution of flexibility differs for our three bnAb lineages. Absolute classical quasi-harmonic entropy for the CG antibody model of the heavy (**A**) and light (**B**) chains, shown as the contribution to the absolute free energy ($-TS$). The absolute entropy magnitude tends to be larger for the heavy chains than for the light chains because they have more amino acids (e.g. 123 aa. for 3BNC60 HC vs. 98 aa. for 3BNC60 LC).
DOI: https://doi.org/10.7554/eLife.33038.008

is reflected in both the RMS fluctuations and the conformational entropy (*Figure 1B* and *Figure 3A*). This result indicates that increased flexibility is not always required to achieve breadth, even for antibody lineages that target the same region (the CD4 binding site). In fact, (*Fera et al., 2014*) attribute the breadth increase in the CH103 lineage to a reorientation of the LC relative to the HC, which alleviates steric clashes with gp120 variants that have longer V5 loops. Furthermore, the variation in flexibility that we report for the CH103 lineage is consistent with earlier studies of non-broadly neutralizing antibodies, which also showed a loss of flexibility during maturation (*Chong et al., 1999*; *Thorpe and Brooks, 2007*; *Wong et al., 2011*), and with the traditional paradigm of affinity maturation (*Foote and Milstein, 1994*; *Wedemayer et al., 1997*). The light chains of the CH103 also show a loss of flexibility upon maturation, although this change is more modest.

For the PGT121 lineage, our MD simulation results exhibit greater complexity. For the heavy chains of this lineage, flexibility changes are observed in the CDR3 region (*Figure 1C*). In this antibody, CDR3 has a 11 residue insertion relative to the 3BNC60 and CH103 lineages, and is involved in binding to gp120 glycans (*Garces et al., 2015*). The CDR3 heavy chain compared to the germline loses flexibility with maturation, again consistent with the traditional paradigm. However, our simulations did not show statistically significant flexibility differences outside of the CDR3 (as indicated by 95% confidence limits associated with 2 × Std. Dev.) suggesting that the overall flexibility reduction of PGT121 HC upon maturation is rather modest. Differences in the conformational entropy for this antibody (*Figure 3*) are consistent with this observation. Although the flexibility differences for the HC of the PGT121 lineage are modest, the variation in flexibility for the LC is significant. We studied three structures – the mature PGT121 antibody, the germline structure, and an antibody construct in which a lightly-mutated heavy chain (3 hr with 11% amino acid mutations, [*Sok et al., 2013*]) was paired with a heavily-mutated light chain (109L with 34% mutations, [*Sok et al., 2013*]). Since the 109L LC is heavily mutated (i.e., like the mature PGT121 LC, which is 28% mutated [*Sok et al., 2013*]), the 109L LC can be considered another variant of the mature light chain. Since it is paired with a lightly mutated heavy chain, we consider the 3 hr/109L antibody structure to be representative of an intermediate stage of PGT121 evolution. While the mature variant is less flexible than the germline in the regions FWR3-FWR4, the intermediate construct has high flexibility, particularly in the framework regions (see *Figure 1F* and *Figure 2B*). This result underscores that the dramatic flexibility increase observed for the 3BNC60 HC above is not an isolated occurrence. Specifically, both chains of the 3H109L PGT121 intermediate have higher entropies than the corresponding germline and mature structures, with the increase being more pronounced for the light chain (*Figure 3*), on which we focus below.

To understand the physical basis for the markedly increased fluctuations in the 3 hr/109L intermediate, we examined the simulated structures and the sequences of the mature PGT121 and of the

intermediate 3 hr/109L. In the PGT121 light chain, the FWR3 region near residue 68 is well ordered, and interacts with the CDR1 region near residue S27 (*Figure 2—figure supplement 1*). Hydrogen bonds are formed between S27 (both, side chain and backbone oxygens) and G68 (backbone amide). In the 109L LC the serine is an alanine, and the stabilizing hydrogen bonds do not form, which allows the FWR3 region to unfold locally near the three residue insertion 66(a)P-66(b)D-66(c)I. In the mature PGT121 light chain, this insertion is mutated to 66(a)D-66(b)S-66(c)P (*Figure 2—figure supplement 1*), and in conjunction with A27S, results in a stable conformation. However, it is possible that this conformation is only weakly stable, such that minor changes in the adjacent structure (such as the S27A mutation in the CDR1) could cause unfolding. Therefore, the stability of FWR3 in the intermediate 109L LC appears to depend, at least in part, on the sequence identity of CDR1, which might seem surprising because the framework regions are thought to provide a relatively rigid scaffold to which variable and flexible CDR regions are attached. Other sequence differences between the mature LC and 109L in or near the insertion region could also be significant, such as S65T or S66(c)I. However, examination of the trajectory did not implicate these residues in the instability directly, because they are oriented away from the CDR1 loop. These results suggest that the complete explanation for the increased flexibility is quite complicated.

Overall, our results demonstrate that flexibility throughout the antibody can be changed significantly by mutations in both the FWR and CDR regions, and, more generally, that affinity maturation in bnAbs does not follow the rigidification paradigm in all cases.

## A model of AM suggests that the binding affinity of the germline BCR to the conserved regions of the epitope determines the role of framework mutations on bnAb evolution

Our analyses of MD simulation results clearly indicate significant 'exceptions' to the paradigm of antibody rigidification during affinity maturation. For the 3BNC60 lineage, flexibility progressively increased, while the opposite is true for the CH103 lineage. For the PGT121 lineage, our results indicate that the intermediate is more flexible than either the mature or germline antibodies, so that this is an example of a more complex maturation trajectory. The germline for the CH103 lineage bound strongly to the CD4 binding site of the founder HIV envelope glycoprotein (*Liao et al., 2013*). Conversely, ancestors of 3BNC60 show lower potency and breadth of neutralization (*Scheid et al., 2011*; *Klein et al., 2013*). Therefore, although the founder viral strain for the patient from whom 3BNC60 was harvested was not available, this result suggests that the 3BNC60 ancestors bind weakly to the CD4 binding site of the epitope that induced bnAb evolution. In general, it is not likely that a naive germline B cell will have a strong affinity for the shared conserved residues. Thus, we hypothesized that the traditional paradigm of reduced flexibility with maturation applies when the germline BCR binds strongly to the conserved shared residues of the variant antigens that induce the evolution of bnAbs. More complex evolution of flexibility can result when this is not the case, as we found for the 3BNC60 and PGT121 lineages. To explore this hypothesis in light of processes that occur during AM in the presence of variant antigens, we developed a simplified model of AM.

### Description of the affinity maturation model

Past studies analyzing the evolution of bnAbs in infected persons show that often diversification of viral strains precedes the development of bnAbs (*Liao et al., 2013*; *Doria-Rose et al., 2014*; *Bhiman et al., 2015*). Thus, we studied AM induced by a cocktail of multiple variant antigens. Our computational model of affinity maturation of B cells includes somatic hypermutation (SHM) in the CDR as well as in the FWR. The purpose of our computational model is not the quantitative reproduction of existing experiments or our MD simulation results, but rather to provide mechanistic insights into how mutations in the CDR and FWR regions change the flexibility of Ab structure to influence the development of breadth and potency. As in recent models of the mutation and selection phenomena that occur during AM induced by multiple variant antigens, which predicted phenomena that were experimentally validated (*Wang et al., 2015*), our model for AM does not explicitly consider the motion of B cells in space or the atomistically detailed structure of BCRs and antigens. The model is conceptually similar to the one described by (*Wang et al., 2015*) and (*Shaffer et al., 2016*), but we adopt a different approach to modeling the BCRs, the variant antigens, and their interactions with each other. The present approach explicitly considers the flexibility

of the antibody structures in determining the free energy of interactions between BCRs and antigens. We first describe how the binding free energy is estimated, and then show how the various steps of AM are computed (further details are provided in the Materials and methods).

The variant antigens have a certain fraction $\lambda$ of their amino acid sequence that is the same. This fraction represents the shared conserved residues of the targeted epitope; $(1-\lambda)$ is the fraction of amino acids that differ. Each B cell is characterized by its binding energy $E_c$ to the shared conserved region, and its binding energy $E_{vi}$ to the variable part of each antigen $i$. Lower (i.e., more negative) binding energies indicate stronger binding. Further, the flexibility of the BCR can play an important role in determining the entropy loss on binding. For more rigid BCR structures, there is a smaller loss of entropy upon binding; for more flexible BCRs, the binding free energy is less favorable (for the same binding energy) because of the greater entropy loss. However, since more flexible structures can bind to more diverse antigen variants, increased flexibility can contribute favorably to the binding energy. For simplicity, we characterize the structural flexibility of a BCR by a single parameter $Q$, which take values between 0 and 1; $Q = 0$ and $Q = 1$ correspond to a highly flexible and a highly rigid structure, respectively. Given the analysis above, we introduce an approximate model for the binding free energy, for a BCR characterized by $Q$, $E_c$ and $E_{vi}$ (defined above) with antigen variant, $i$, it is defined as:

$$E = Q(\lambda E_c + (1-\lambda)E_{vi}) + (1-Q)E_0 \tag{1a}$$
$$= QE_w + (1-Q)E_0, \tag{1b}$$

$E$ becomes more favorable as mutations are acquired that increase the binding energy of the BCR to the residues of the antigen. The first term in *Equation 1*(b), $QE_w$, includes an approximation to the entropy contribution to the free energy, based on the reasonable assumption that the effect of flexibility can be captured by scaling down the energy $E_w$. As the BCR becomes more flexible (smaller $Q$), because of the greater entropy loss associated with binding, the net binding free energy ($QE_w$) is smaller for the same value of $E_w$. $E_0$ corresponds to a moderate favorable 'generic' binding free energy to all variant antigens accessible through conformational plasticity. The second term in *Equation 1*(b) plays an increasingly important role for flexible BCR structures (smaller values of $Q$). For simplicity, we assume that mutations in the CDR affect the binding energy ($E_c$ or $E_{vi}$, depending on their location) and that mutations in the FWR only affect the rigidity of the BCR ($Q$) and not the binding energy. The model could be refined by taking into account different flexibilities of the epitopes on variant antigens. As there is no way to know the distribution of epitope flexibilities, we have instead studied the influence of varying the value of $E_0$.

More negative binding free energies correspond to stronger affinities. The absolute value of the binding free energy is arbitrary as free energies are determined only up to an additive constant. We assume that a binding free energy of zero is the minimum required for antigen binding (for example only B cells that bind to antigen with a more favorable binding free energy than zero can seed a Germinal Center (GC)). The binding free energies in our model are expressed in units of $k_BT$ (see text following *Equation 4* in Materials and methods).

We simulate the processes that occur during AM using a stochastic model, as is appropriate for an evolutionary process. The set of rules that define AM are derived from experimental studies of affinity maturation with a single antigen (*Allen et al., 2007*; *Berek and Milstein, 1987*; *Tas et al., 2016*; *Victora and Nussenzweig, 2012*), and these serve as instructions that are executed by the computer. The following steps are executed for the situation where there exists a number of variant antigens displayed on the Follicular Dendritic Cells (FDCs) in the GC.

## Seeding of a GC and somatic hypermutation (SHM)

We select a few founder B cells that bind to one of the variant antigens with a binding free energy above a threshold (defined above) to seed a GC. The founder B cells expand reaching a population of 1,536 cells. Then, the AID gene turns on and mutations are introduced with a probability determined by experiments: each B cell of the dark zone divides twice per GC cycle (four divisions per day) (*Zhang and Shakhnovich, 2010*) and a mutation occurs with a probability of 0.20 per sequence per division (*Berek and Milstein, 1987*). The mutations affect both the CDR and the FWR. Although the FWR constitutes over two thirds of the variable domain by sequence, mutational hotspots have mostly been found in the CDR (*Neuberger and Milstein, 1995*; *Tomlinson et al., 1996*;

*Wagner and Neuberger, 1996*). Thus, we assume that the probability that a mutation occurs is higher in the CDR ($p_{CDR}$) than in the framework ($p_{FWR}$). The precise values of the parameters used are provided in *Supplementary file 1* and *2*.

## CDR mutations

CDR mutations can cause a B cell to undergo apoptosis (for example, by making the BCR unable to fold), be silent (e.g., synonymous mutation), or modify the binding energy. The probability that a CDR mutation will follow one of these paths is governed by probabilities obtained from experiments (*Berek and Milstein, 1987*). Apoptosis occurs half of the time, 30% of mutations are silent, and 20% are binding energy-affecting mutations (*Zhang and Shakhnovich, 2010*). Experimental studies of protein-protein interactions indicate that binding energy-affecting mutations are more likely to be deleterious than advantageous (*Moal and Fernández-Recio, 2012*). Therefore, in our model, the change in binding energy is sampled from a shifted log-normal distribution whose parameters are chosen to approximate the observed empirical distribution of changes in binding energies upon mutation. In our model, the CDR mutations change $E_c$ or $E_{vi}$, depending on their location, but do not affect flexibility ($Q$).

## FWR mutations

In the case of a mutation in the FWR, we assume that the likelihood of undergoing apoptosis is very high (80%) to account for the role of the FWR in maintaining the structural integrity of antibodies. Non-lethal mutation yields a change in the flexibility of the structure, which is represented by a change in the rigidity parameter $Q$ that is randomly sampled from a Gaussian distribution. We allow $Q$ to vary between a state of maximum rigidity ($Q = 1$) and a state of high flexibility ($Q = 0.1$). If a mutation would increase $Q$ above one or decrease it below 0.1, we simply set $Q$ equal to the boundary value of 1 or 0.1, respectively. Because FWR regions tend to be farther from the binding interface than CDR regions, we assume that FWR mutations do not directly affect binding energy ($E_c$ or $E_v$); their effects are introduced indirectly through changes in $Q$.

## Selection

After SHM, the mutated B cells then migrate to the light zone of the GC, where selection takes place through competition for binding to antigens displayed on FDCs and for receiving T-cell help. We do not model the spatial migration step explicitly, but rather selection occurs after the rounds of division and somatic hypermutation in the dark zone. As our goal is to explore how the qualitative nature of the evolutionary paths followed by B cells in the GC depend upon the strength of binding of the germline B cells to the conserved residues of the epitope, treating migration explicitly is not necessary.

In the GC, B cells interact with antigen displayed on FDCs. B cells with receptors that bind more strongly to the antigen will likely internalize more antigen. They are thus more likely to display larger numbers of antigen-derived peptides bound to MHC Class II molecules on their surface. The B cells that internalize antigen and display peptide-MHC molecules on their surface compete with each other for the limited number of T helper cells in the GC. If there is a productive interaction of peptide-MHC molecules on B cells undergoing selection and the T cell receptor on the surface of T helper cells, the B cell receives a survival signal, and if it does not, apoptosis results (*Foy et al., 1994*; *Crotty, 2015*). B cells that display more peptide-MHC molecules on their surface are more likely to receive this survival signal.

We model the two-step selection process noted above as follows. First, each B cell successfully internalizes the antigen it encounters with a probability that grows with the binding free energy and then saturates, following a Langmuir form (see *Equation 4* in Materials and methods). B cells that successfully internalize antigen can then go on to the second step, while the others die automatically. The B cells that internalize antigen are ranked according to their binding free energy, which is expected to correlate with the concentration of pMHC that they display to T cells, and only the best performers are selected (see *Supplementary file 1* and *2* for details on parameter values).

In the presence of multiple antigen variants present on FDCs, a B cell could interact with and internalize several variants at a time or just one variant. However, there is no experimental data to describe how heterogeneously different antigens are distributed on the surface of FDCs at a given

time. It was shown previously that, if B cells interact with multiple variants of the antigen in every cycle, bnAbs are less likely to evolve (*Wang et al., 2015*; *Shaffer et al., 2016*). Therefore, in our model we assume that a B cell encounters only one antigen variant on FDCs at a time; a different choice would lower the probability of bnAb evolution. The encountered antigen variant is chosen randomly with a uniform probability for each B cell during each round of selection.

## Recycling, exit for differentiation, and termination of the GC reaction

Most (70%) B cells that are positively selected are recycled for further rounds of mutation-selection while a few randomly selected B cells exit the GC to mimic the differentiation into memory and antibody-producing plasma cells (*Oprea and Perelson, 1997*). The GC reaction comes to an end in three possible circumstances: 1] All the B cells die, thus extinguishing the GC. 2] If the number of B cells in the GC exceeds 1,536 cells; this is a proxy for the antigen being consumed by the B cells 3] When the number of cycles, or time, exceeds a maximum number (250 cycles, or $\approx$ 125 days assuming two cycles per day; see *Supplementary file 1* and *2* for full parameter values), which is a proxy for the antigen having decayed or loss of antigen from the GC.

Although the model that we simulate lacks structural detail, it reproduces experimentally observed order-of-magnitude increases in the binding affinity during AM induced by a single antigen (see Materials and methods and *Figure 4—figure supplement 1*).

## Simulated AM trajectories

For each simulated AM trajectory in the GC, we record the total number of B cells over time as well as the number of acquired energy-affecting CDR mutations and rigidity-affecting FWR mutations. Due to the stochastic nature of B cell evolution during affinity maturation, we carried out $10^4$ simulations for each set of conditions studied, and combined the results to obtain meaningful statistics. Our results thus reflect the probability with which certain types of evolutionary patterns would be observed for each set of conditions that we studied. Because mutation probabilities are less than unity, cell division can generate cells with identical values for binding energies and rigidity. Thus, the GC contains sets, also called clones, of these functionally identical B cells. We analyze the properties of the clone with the largest number of cells at the end of the AM trajectory, and consider this to be representative of the properties of most of the antibodies generated. We also trace the evolutionary trajectories of these clones to determine the history of mutations in the CDR and FWR regions that shaped these final properties. To analyze the antibody breadth, we compute the binding free energy of this largest clone to a panel of 100 antigens different from those against which the antibody matured. For this purpose, we take the overlap parameter $\lambda$ and the binding strength with the conserved region to be the same as in the simulations of affinity maturation, but the binding strength with the variable region is randomly selected for each new antigen in the panel (Materials and methods). As a proxy for breadth, we compute the median binding free energy with panel antigens of each antibody over the maturation pathway . This allows us to track how breadth evolves during the course of affinity maturation. The higher the magnitude of the median binding free energy the greater the cross-reactivity of the Abs to variant antigens.

The results described in the main text have been carried out with ten variant antigens that share 90% sequence identity ($\lambda$ = 0.9), and three B cells seeding the GC. The precise number of antigens and the value of $\lambda$ used in the simulation are not important for the qualitative results that we present (for example, see Materials and methods and *Figure 4—figure supplement 2* where results with five antigens are shown). Similarly, increasing the number of seeding B cells to seven did not affect our results.

## Results of the affinity maturation model

To explore the hypothesis suggested by our MD simulation results, we considered two classes of activated germline B cells that seed the GC upon immunization with a cocktail of variant antigens: 1) those with a high favorable value of the binding energy to the shared conserved region of the antigens, $E_c$ (such as those that have a long finger-like structure that binds to the CD4 binding site in the VRC01 family of antibodies); and 2) those with a weak/moderate binding energy to the shared conserved regions, which are expected to be more common upon natural infection, since most naïve B cells are unlikely to have strong interactions with the conserved residues prior to affinity

maturation. The Abs start with a high value of the rigidity parameter, $Q = 0.8$. The exact initial value of $Q$ had little qualitative effect on the outcome (data not shown).

Our results show strikingly different Ab evolution patterns and outcomes depending on the initial value of $E_c$ (*Figure 4*). When the germline B cells bind to the conserved residues with relatively high affinity (e.g., $E_c \approx -4$), B cells evolve CDR and FWR mutations that strengthen the binding energy ($E_c$) to the shared conserved residues and reduce the flexibility of the antibody (increase $Q$); if the initial $Q$ is already high, the increase may be relatively small. This is consistent with our MD simulation results for the CH103 lineage (and the traditional paradigm). The resulting antibodies are both broad and potent as reflected by the median binding free energy to the panel of antigens and the absolute magnitude of $E_c$ (*Figure 4*). Thus, these antibodies acquire breadth and potency by focusing their interactions with the variant antigens on the shared conserved residues (more favorable $E_c$). This is the situation with the class of antibodies that acquire breadth and potency by focusing interactions with the CD4 binding site.

In the case where the binding energy of the germline B cell to the shared conserved residues is weaker (taken to be $E_c = 0$), we find that B cells typically need to acquire FWR mutations that increase their flexibility in order to be able to be positively selected in the presence of multiple variant antigens (*Figure 4*). This is made evident by the fact that in simulations where changes to flexibility are disallowed, GC reactions are 35% less likely to complete successfully because all the B cells undergo apoptosis as AM ensues (data not shown). Akin to enzymes in a changing environment of substrates (*Raman et al., 2016*) discussed earlier, the FWR mutations enable the B cells to bind loosely to the diverse variant antigens that act as the selection forces during affinity maturation. Since the value of $E_c$ during the early stages of affinity maturation is not high, one way for the B cells to bind sufficiently strongly to diverse variant antigens and be positively selected is by evolving FWR mutations that increase conformational flexibility. Upon becoming more flexible and thus surviving selection against the variant antigens, in subsequent rounds of mutation and selection, the B cells can start to acquire CDR mutations that increase the binding energy to the conserved residues, thus acquiring breadth. Most bnAbs harvested at this stage of evolution will exhibit high breadth and relatively low potency because the free energy of binding to each variant antigen is not very high. This is because the flexibility of the BCR/antibody implies that a high entropic penalty has to be paid upon binding to the antigens (also see *Equation 1* in the limit of small $Q$). In some cases, however, the flexible bnAbs can also be highly potent. This is because, as our calculations show, some flexible bnAbs that are produced at this stage of AM can develop strong interactions with the conserved residues ($E_c \approx -11$). This may be the case for the 3BNC60 antibody, which was isolated from patients with the specific selection criteria of high breadth and potency. As our MD simulation results show, for this lineage, flexibility progressively increases with maturity; 3BNC60 exhibits high breadth, with the potency expected to be comparable to that of CH103. Although we could not find specific comparative data for 3BNC60, another 3B class of bnAb, 3BNC117, is slightly more potent than CH103 (*Chuang et al., 2013*; *Eroshkin et al., 2014*). Indirect comparisons based on available data (*Scheid et al., 2011*; *Chuang et al., 2013*) suggest that 3BNC55, another bnAb with the P61 FWR mutation is less potent than CH103. This may be because it has not yet acquired the mutations required for strong interactions with the conserved residues, and our calculations (*Figure 4*) suggest that its flexibility makes 3BNC55 less potent.

*Figure 4A* shows that, in the case where germline B cells have a weak binding energy $E_c$, if the evolutionary trajectories progress beyond a reasonably favorable value of $E_c$ (which our simulations show to be roughly equal to $E_0$), then framework mutations that make the structure more rigid are acquired much like the case where germline B cells exhibit a stronger $E_c$. Thus, the antibodies become more potent because they do not have to lose conformational entropy upon binding (see also *Equation 1*). Our results show that for the cases where the germline B cells have a low binding energy to the conserved residues, the median binding free energy of the antibodies for the panel of antigens exhibits a broad pattern at generation 400 (*Figure 4B*). Analyses of our results demonstrate that the population that exhibits lower breadth (median binding free energy) and lower final values of $E_c$ is comprised of those antibodies that are more flexible, while those that continue to evolve and become more rigid have a higher breadth and potency (*Figure 4C*). This rigidification, following an initial increase in flexibility that enables the development of sufficiently strong interactions with the conserved residues, would explain why the PGT121 bnAb is more potent than 3BNC60. Also,

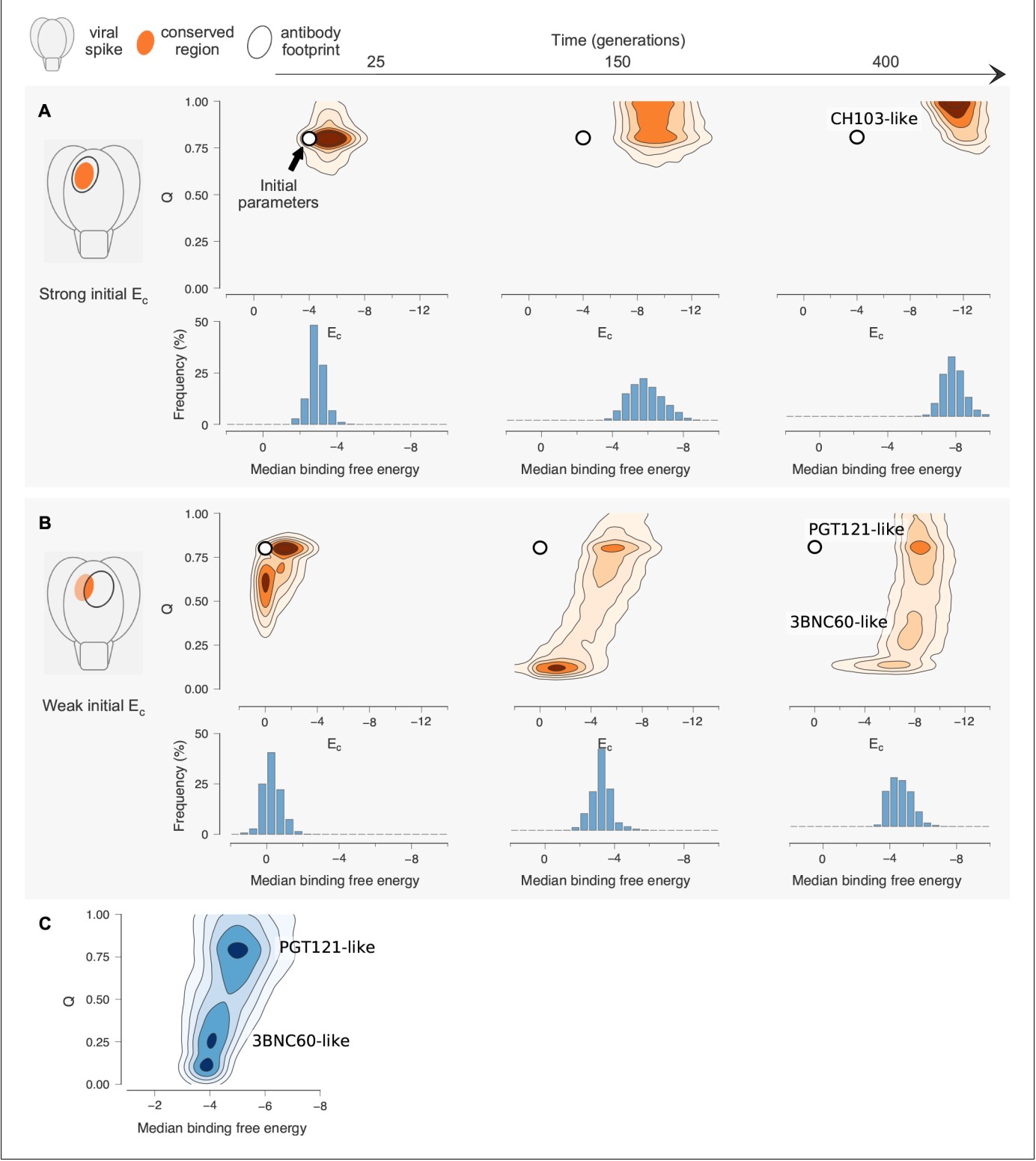

**Figure 4.** Antibody maturation proceeds along alternate routes that depend on the initial binding strength with the conserved region of the antigen. Here we show statistics of the maturation pathways of the largest clones in our affinity maturation simulations. These figures therefore summarize typical features of antibody maturation across many parallel and independent germinal center reactions. (A) Antibody lineages that initially bind strongly with the conserved region of the antigen are likely to accumulate mutations that increase their binding strength and reduce flexibility. Starting parameters are indicated with an arrow. In our simulations, mutations in the CDR affect binding energies directly, while mutations in the FWR affect flexibility. Typical binding free energies with panel antigens, a proxy for breadth, strengthen steadily over the course of maturation (represented for

*Figure 4 continued on next page*

*Figure 4 continued*

generations 25, 150 and 400). (B) Antibodies that initially bind weakly with the conserved region typically become more flexible while increasing their binding strength. Such antibodies may subsequently begin to rigidify as they mature. The typical binding free energies with panel antigens increase slightly faster than in the strong conserved binding case above, but final binding free energies are not as strong. (C) The most potent antibodies at the end of the maturation process (generation 400), measured by median binding energy with panel antigens, are those that are the least flexible.

DOI: https://doi.org/10.7554/eLife.33038.009

The following figure supplements are available for figure 4:

**Figure supplement 1.** Simulations of affinity maturation against a single antigen agree with experimental results.
DOI: https://doi.org/10.7554/eLife.33038.010

**Figure supplement 2.** Similar values of the overlap parameter lead to qualitatively similar results.
DOI: https://doi.org/10.7554/eLife.33038.011

**Figure supplement 3.** Shifting $E_0$ affects the preferred antibody maturation pathway.
DOI: https://doi.org/10.7554/eLife.33038.012

these results are consistent with our MD simulations, which show that the intermediate is more flexible than both the germline or the mature antibodies.

For cases where the germline B cells exhibit a weak binding energy for the shared conserved residues of the antigens, we also observe some evolutionary trajectories where mutations quickly arise in the CDR regions that increase the affinity to the conserved residues and this is followed by the antibody structures evolving to become even more rigid. Such cases are less common.

The AM simulation results described above were obtained with a particular value of the "generic" binding energy $E_0$ (nonspecific binding energy accessible to flexible antibodies; see *Equation 1 a,b*). We have explored the effects of varying the value of this parameter on our results when the germline B cell does not bind strongly to the conserved residues of the epitope (weak initial $E_c$). As *Figure 4— figure supplement 3* shows, shifting $E_0$ to very strong or weak binding energies affects the preferred antibody maturation pathway. A significantly more favorable value of $E_0$ than that used to obtain the results in *Figure 4* makes flexibility-increasing framework mutations more beneficial, and thus drives antibody evolution more along 3BNC60-like trajectories (compare *Figure 4—figure supplement 3* with *Figure 4B*). In this case, antibodies begin to rigidify and become more like the PGT121 lineage later in the maturation cycle, after mutations that make $E_c$ (binding energy to the conserved residues) very strong have been acquired. So, if $E_0$ is stronger than what was used to obtain the results in *Figure 4*, the qualitative behavior does not change, but the 3BNC60-like antibodies are more likely to be observed for a greater duration of the affinity maturation cycle and are likely to be more potent. If $E_0$ is weaker than what was used to obtain the results in *Figure 4*, and we use the same antigen concentration as before, most of the B cells die and the GC is likely to be extinguished. This is because affinity maturation is frustrated when there are multiple variant antigens and the initial $E_c$ is weak, as has been described before (*Wang et al., 2015*; *Shaffer et al., 2016*), and a path to relieving this frustration by evolving flexibility through FWR mutations is no longer available (because of weak $E_0$). In this circumstance, if the antigen concentration is increased, we find that the GCs are not extinguished with high probability, and more CH103-like maturation trajectories are promoted (compare *Figure 4—figure supplement 3* with *Figure 4A*). This is because CDR mutations that improve binding with the conserved region (stronger $E_c$) become relatively more beneficial early in the maturation process since $E_0$ is weak and there is little benefit to evolving FWR mutations that affect flexibility. Consistent with the points noted above, varying values of $E_0$ also change the rate of acquiring CDR versus FWR mutations (*Figure 6—figure supplement 1*), with stronger values of $E_0$ promoting FWR mutations more than in Figure 6B and weaker values of $E_0$ suppressing FWR mutations.

## Discussion

HIV is a highly mutable pathogen that continues to rage in many parts of the world. A promising strategy for the development of a prophylactic vaccine against such pathogens is to design immunization protocols and immunogens that can induce antibodies (bnAbs) that can neutralize diverse circulating mutant strains of the virus. BnAbs that evolve naturally in infected persons exhibit high levels of CDR and FWR mutations. In some instances, FWR mutations are important for breadth

(*Scheid et al., 2011*) and, in at least one case, FWR mutations have been shown to increase antibody flexibility (*Scheid et al., 2011*; *Klein et al., 2013*). The standard paradigm, based on the development of strain-specific antibodies is that FWR mutations result in a more rigid antibody structure (*Schmidt et al., 2013*; *Eisen and Chakraborty, 2010*; *Foote and Milstein, 1994*; *Wedemayer et al., 1997*; *Thorpe and Brooks, 2007*). Since bnAbs develop by AM in the presence of diverse mutant strains, we explored the hypothesis that FWR mutations increase the number of accessible conformational states of the antigen binding region, thus enabling binding to the variant antigens. This, in turn, may allow B cells to avoid frequent apoptosis due to the conflicting selection forces represented by the variant antigens.

To test this hypothesis, we first performed atomistically detailed molecular dynamics simulations of the structures of antibodies at three different time points during the maturation process for three bnAb lineages (3BNC60, PGT121, and CH103). Only the 3BNC60 lineage, showed the continuous increase in conformational flexibility previously reported (*Scheid et al., 2011*; *Klein et al., 2013*). Structures in the CH103 lineage progressively became more rigid with maturation, consistent with the standard paradigm. For PGT121, the germline and mature forms had similar levels of conformational rigidity, while an intermediate construct was more flexible, suggesting a more complex maturation process.

To interpret these complex results in light of the evolutionary forces at play during the development of bnAbs by AM, we developed and studied a simple computational model of AM driven by multiple variant antigens. The essential findings from our simulations of affinity maturation are depicted schematically in *Figure 5*, which illustrates how the three different cases studied here evolve during AM; details are given with *Equation (1)* and the discussion of *Figure 4* above. Specifically, *Figure 5* shows the change in rigidity ($Q$) and the binding energy to conserved residues ($E_c$) as a function of the value of $E_c$ for the germline B cell. If $E_c$ is sufficiently favorable ($E_c \ll 0$), as in the CH1 E103 case (*Figure 5*, orange), AM follows the classical path of increasing rigidity. If $E_c$ is low ($E_c \approx 0$), as is likely the case for 3BNC60 (green) and PGT121 (purple), rigidity initially decreases via mutations in the FWR regions. The increase in flexibility allows the antibodies to bind a broader set of antigens via conformational plasticity, with a relatively low but sufficiently strong affinity, to permit them to survive until they acquire additional mutations that enhance $E_c$. That is, a more negative (or stronger) binding free energy, $E$, can be acquired by increasing the magnitude of the second term in *Equation 1*b at the expense of the first term. Then, further beneficial CDR mutations that strengthen $E_c$ can follow, thus making $E_w$ more negative (stronger binding) for diverse variant antigens. This scenario is consistent with both 3BNC60 and PGT121 cases. Most antibodies harvested at this stage of AM will not be very potent because flexible antibodies lose more entropy upon binding, thus limiting the value of $E$ (green curve); however, some antibodies, such as 3BNC60, can also develop a highly favorable value of $E_c$. At this stage in the AM process, rigidification can increase the antibody potency further. This is observed in our simulations once $E_c$ is sufficiently high that $E_w < E_0$; conformational plasticity is now no longer required and the selection pressure will increase $Q$ to enhance the effect of the first term in *Equation 1*b. This phenomenon is consistent with the AM trajectories of the PGT121 lineage (purple curve). The reason that PGT121 is more potent than 3BNC60 is likely that it has proceeded further along the evolutionary trajectory and become more rigid.

In our model, we observe that the accumulation of mutations in the CDR and FWR regions differs according to the maturation pathway (*Figure 6*). Antibodies that start with a relatively strong binding with the conserved region (CH103-like) accumulate mostly affinity-improving CDR mutations while some FWR mutations contribute to rigidity (*Figure 6A*). In contrast, antibodies that start with weaker binding to the conserved region (PGT121- and 3BNC60-like) more rapidly accumulate FWR mutations that improve flexibility during the early stages of AM (*Figure 6B*). This is a prediction of our model which is supported by experimental results from (*Klein et al., 2013*) where a phylogenetic analysis of antibodies from the 3BNC60 lineage places two important FWR mutations near the root, suggesting that they emerged early during the developmental process. Later CDR mutations then contribute to increased binding affinity. Our predictions could be further tested by examining the relative rates at which CDR and FWR mutations are acquired during the evolution of the CH103 and PGT121 bnAb lineages.

The results have an important implication for the design of immunization strategies to induce bnAbs. If germline B cells can be activated that bind sufficiently strongly to the shared conserved residues of a target epitope of a class of bnAbs (e.g., the CD4 binding site), there is no need for the

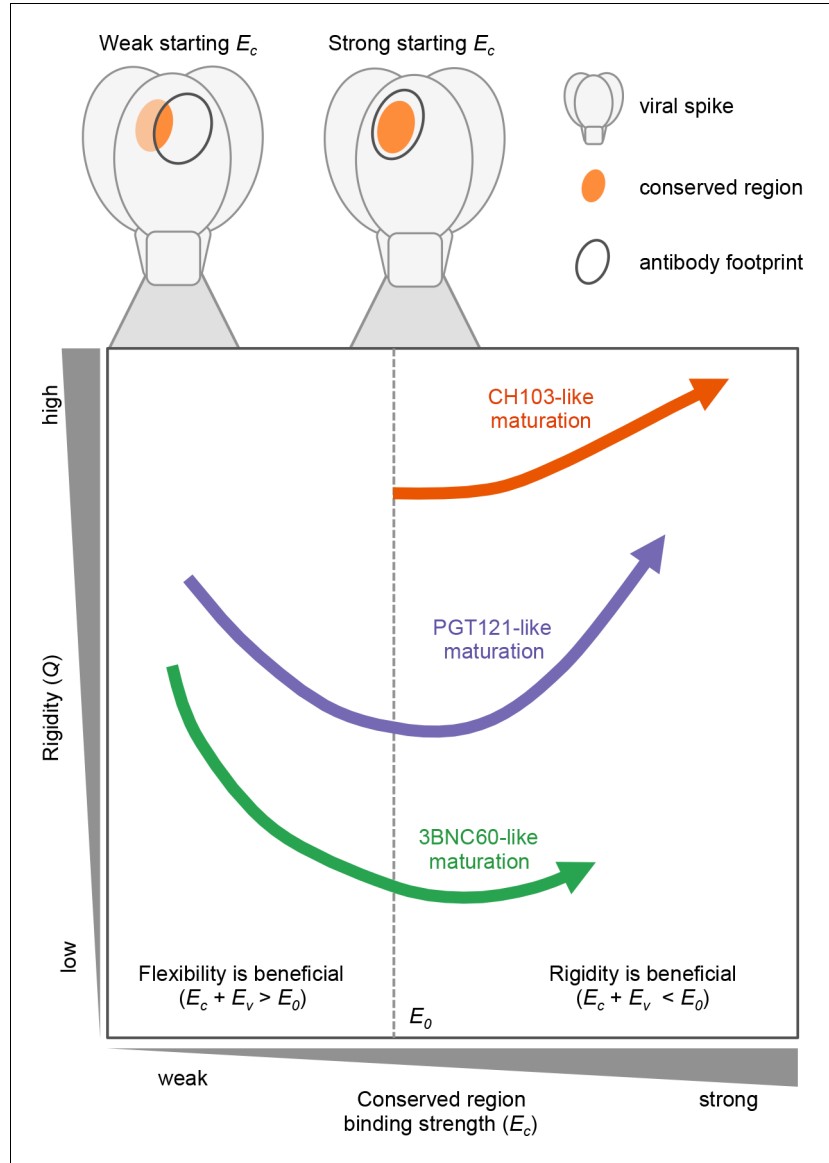

**Figure 5.** The three bnAb lineages we studied evolve through different paths that depend on the binding strength of their germline to conserved epitopes. CH103 (in orange) has a strong starting binding energy for the conserved residues, $E_c$. It follows the traditional evolution pathway and quickly rigidifies while enhancing its $E_c$. PGT121 (in purple) and 3BNC60 (in green) have germlines with a weaker $E_c$. To survive selection during affinity maturation with multiple different antigens, they follow the same pathway as some enzymes (*Raman et al., 2016*) in changing environments: they first become m E1ore flexible which allows them to bind all antigens with limited potency; later they acquire mutations that enhance $E_c$ and increases their binding potency.
DOI: https://doi.org/10.7554/eLife.33038.013

subsequent immunization with variant immunogens to induce FWR mutations that increase flexibility of the BCRs as AM ensues. Thus, our results suggest that strategies that result in inducing such germline B cells (*Steichen et al., 2016*; *Escolano et al., 2016*; *Jardine et al., 2016*; *Jardine et al., 2015*; *Jardine et al., 2013*) would considerably simplify the design of boosting immunogens.

Our study represents a first step in understanding the complex synergies between FWR and CDR mutations that are in play during the evolution of bnAbs. We hope our findings encourage others to further study this fundamental problem at the intersection of immunology and evolutionary biology with important implications for health. Several caveats in our study are worth noting as they provide suggestions for such studies. We were able to study only three bnAb lineages because of the lack of

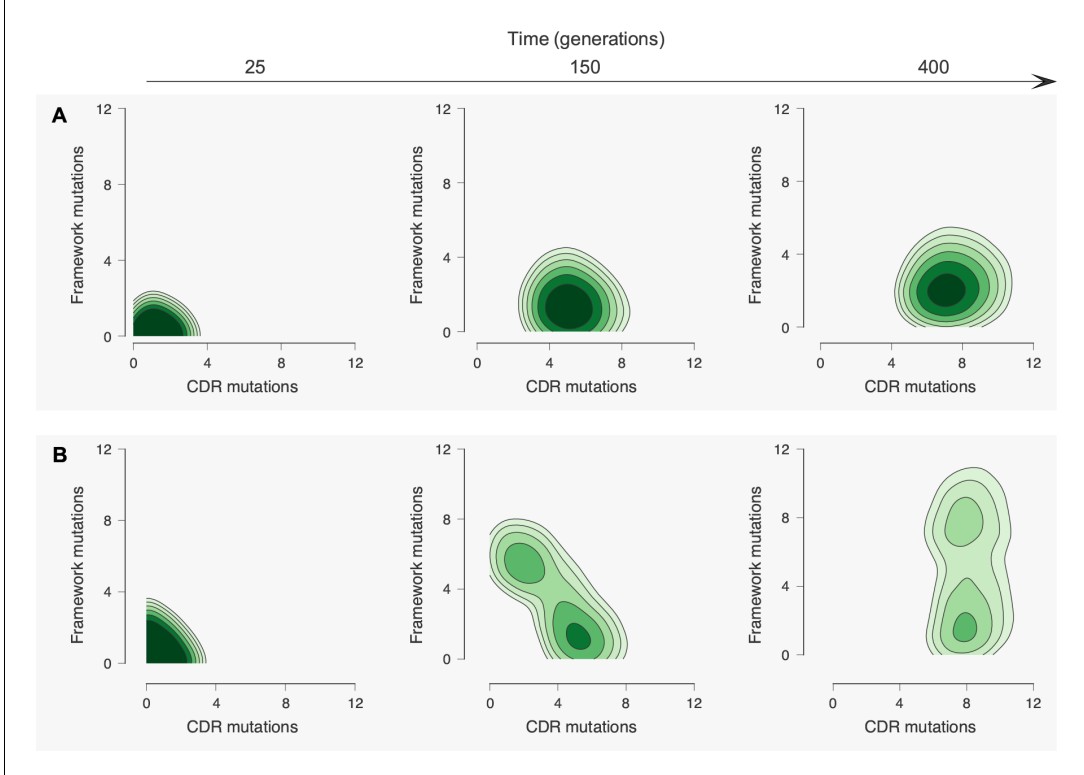

**Figure 6.** Accumulation of mutations in the CDR and framework regions varies according to the maturation pathway. (**A**) Antibodies that have a strong initial binding energy for the conserved residues, $E_c$, tend to first accumulate CDR mutations. Later, these antibodies rigidify through framework mutations. (**B**) Antibodies that have germlines with a weaker starting $E_c$ are more likely to acquire early framework mutations to increase flexibility, improving the odds of surviving selection during affinity maturation with multiple antigens. Note that nearly all lineages possess early framework mutations, in contrast to the maturation trajectories in (**A**).

DOI: https://doi.org/10.7554/eLife.33038.014

The following figure supplement is available for figure 6:

**Figure supplement 1.** Shifting $E_0$ affects the accumulation of mutations in the CDR and framework regions along the maturation pathway.

DOI: https://doi.org/10.7554/eLife.33038.015

high resolution antibody structures at different points of maturation. Furthermore, we could only study three time points in the evolutionary pathway of each lineage, and for PGT121, the structure we studied is a chimeric construct and not a true intermediate. Our results suggest that complex evolutionary trajectories are followed under different conditions, and structural analysis of various bnAb lineages at multiple time points during their maturation pathway could further test our predictions and reveal new phenomena. For example, studies that provide a finer temporal resolution during AM could reveal additional types of evolutionary trajectories not described by our current results. Iteration between such experiments and development of more refined models would further elucidate the complex processes that occur when AM results in the development of bnAbs.

## Materials and methods

Data and some analysis for molecular dynamics simulations are available in a GitHub repository (*Barton, 2018*). A copy is archived at https://github.com/elifesciences-publications/paper-bnAb-flexibility.

### Molecular dynamics simulations

Antibody structures listed in *Table 1* were downloaded from the PDB databank, and include both the variable regions and the constant regions, which are connected by a single $\beta$-strand for each HC and LC (*Figure 2—figure supplement 2*). To save simulation cost, the structures were truncated to

retain only the variable regions, since the constant regions do not contact the antigens. To truncate all of the antibodies in equivalent locations, a multiple alignment of each chain was performed. The truncation locations correspond to res. 120 and res. 98 for the heavy and light chains of 3BNC60, respectively. Histidine protonation was performed by optimizing visually the hydrogen bonding patterns for each histidine. The CHARMM36 energy function was used for the MD simulations. Each structure was energy-minimized in vacuum using CHARMM (*Brooks et al., 2009*) with restraints on the protein heavy atoms, and immersed in a cubic box of TIP3P water of sufficient size that the smallest distance between the antibody and the box boundary was 11.5 Å. $Na^-$ and $Cl^+$ ions were added to maintain system neutrality, while achieving an approximate salt concentration of 100 mM. The final system sizes were between 40,000 and 51,000 atoms; variations were mainly due to differences in the CDR loop lengths (e.g., the PGT bnAb has the longest CDR3 region, and had about 51,000 atoms). MD simulations were performed using GPU hardware with the program ACEMD (*Harvey et al., 2009*).

Each system was equilibrated for 2ns with restraints on the heavy atoms that were resolved in the ray structure. The Berendsen barostat was used with a relaxation time of 10,000 steps to allow the simulation box to resize according to the new composition. We used a Langevin thermostat to maintain temperature at 298K with a friction constant of $0.1ps^{-1}$. The time-step was 1fs for the first 100ps, and 2fs for the remainder of the equilibration. The non-bonded cutoff was 11.5 Å.

After equilibration, each system was simulated for 100ns in the canonical ensemble. To accelerate the simulations, the non-bonded cutoff was set to 9 Å, the switching function was nonzero at 7.25 Å, long-range electrostatics were evaluated at every other simulation step, and hydrogen masses were increased to 4 a.m.u., which allowed the use of a 4fs time step. For all simulations in which the time-step was larger than 1fs, covalent bonds to hydrogens were constrained using SHAKE (*Ryckaert et al., 1977*). To generate statistics for subsequent fluctuation and covariance analyses, the equilibration and production simulations were performed five times starting with different initial velocities.

To ensure that the truncation of the antibody constant regions would not impact the conclusions of this study, we also simulated by MD the 3BNC60 antibodies without truncation (i.e. including both the variable and constant regions, for 400ns for each structure. In these simulations, the constant and variable regions remained well-separated (*Figure 2—figure supplement 2*) suggesting that the interactions between them were relatively weak. Further, the RMSF fluctuations on the basis of these simulations were qualitatively unchanged.

## Analysis of residue fluctuations

Root mean square residue fluctuations (RMSFs) were used as indicators of protein flexibility. To compute RMSFs for each residue, each antibody system was first coarse-grained (CG) to one bead per residue. The coordinate for each bead was taken as the center-of-mass (COM) of the residue. The fluctuation for residue $i$ is computed as

$$\rho_i = \left\langle \left\| \boldsymbol{r}_i^{CG} - \langle \boldsymbol{r}_i^{CG} \rangle \right\|^2 \right\rangle^{\frac{1}{2}}, \tag{2}$$

where $\mathbf{r}_i^{CG}$ is the coordinate triplet of the coarse-grained residue $i$, and brackets indicate temporal averaging. The average and standard deviation was computed for each $i$ from the five simulation repeats. The RMSF calculations were also performed by considering coordinates of the C$\alpha$ atoms only. The results are not shown because they were qualitatively similar.

## Calculation of conformational entropy

Quasi-harmonic entropies of the CG systems were used as a measure of overall flexibility. Mass-weighted covariance matrices were computed for each system, as described by (*Brooks et al., 1995*), for the HC and LC independently. For each system, the five trajectories were concatenated. This is permissible because the covariance analysis is not time-dependent. Both classical and quantum entropies were computed using the corresponding harmonic oscillator expressions (*Andricioaei and Karplus, 2001*). Only classical entropies are shown because the relative differences between classical and quantum entropies were similar.

## Affinity maturation model

Data and analysis for the affinity maturation simulations, including the code used to generate all figures, are available in a GitHub repository (*Barton, 2018*).

## Somatic hypermutation in the dark zone

Then, the AID gene turns on so mutations are introduced with a probability determined by experiments: each B cell of the dark zone divides twice per GC cycle (four divisions per day) (*Zhang and Shakhnovich, 2010*) and mutation appears with a probability of 0.20 per sequence per division (*Berek and Milstein, 1987*). The mutations affect any amino acids in the variable domains, including both the CDR and the FWR. Although the FWR constitutes over two thirds of the variable domain by sequence, mutational hotspots have mostly been found in the CDR (*Neuberger and Milstein, 1995*; *Tomlinson et al., 1996*; *Wagner and Neuberger, 1996*). Thus, we assume that the probability that a mutation occurs is higher in the CDR ($p_{CDR}$ = 0.85) than in the framework ($p_{FWR}$ = 0.15).

## CDR mutations

CDR mutations can cause a B cell to undergo apoptosis (for example, by making the BCR unable to fold), be silent (due to redundancy of the genetic code, i.e. synonymous mutation), or modify the binding energy. The probability that a CDR mutation will follow one of these paths is governed by probabilities obtained from experiments (*Berek and Milstein, 1987*). Hence, apoptosis occurs half of the time, 30% of mutations are silent, and 20% are energy-affecting mutations (*Zhang and Shakhnovich, 2010*). Energy-affecting mutations are likely to be deleterious. Experiments have determined that in protein-protein interactions, between 5% and 10% of the energy-affecting mutations weaken the binding energy (*Moal and Fernández-Recio, 2012*). In our model, the change in binding energy is sampled from a shifted log-normal distribution whose parameters are chosen to approximate the empirical distribution of changes in binding energies upon mutation. This distribution is also used to initialize the binding energies of the founder B cells. We assume that CDR mutations do not affect flexibility ($Q$). Thus, the random *change* in binding energy $\Delta E$ ($E_c$ or $E_v$, depending on the location of the CDR mutation) and the initial variable region binding energies are taken to be

$$\Delta E = \exp(\mu + \sigma r) - o \tag{3}$$

where $r$ is a standard normal variable with mean zero and standard deviation equal to one. Here $o$ is a shift parameter, which is needed to center the log-normal distribution properly with respect to zero, and $\mu$, $\sigma$ are the mean and standard deviation of the log-normal distribution, respectively. These parameters are described in *Supplementary file 1*.

## FWR mutations

As described in main text
Selection in the light zone

After SHM, the mutated B cells then migrate to the light zone of the GC, where selection takes plae through competition for binding to antigens and for receiving T-cell help. B cells with the greatest binding free energy for the antigen presented on FDCs will have a better chance to internalize that antigen, break it down into small peptides, and display it on their MHC molecules. These peptide-MHC (pMHC) complexes can then bind to the cognate receptor of T cells. A productive interaction results in a survival signal to the B cell through CD40-CD40 ligand signaling. B cells that do not receive a survival signal die through apoptosis (*Foy et al., 1994*; *Crotty, 2015*).

We model the biology with a two-step selection process. First, each B cell successfully internalizes the antigen it encounters with a probability that grows with the binding free energy and then saturates, following a Langmuir form. The probability that a B cell successfully internalizes an antigen $i$ that it encounters also depends on the antigen concentration $C_i$. We assume that this probability follows a Langmuir formula:

$$P_{int} = \frac{C_i e^{-E}}{1 + C_i e^{-E}} \tag{4}$$

where E is given in *Equation (1)*. B cells that successfully internalize antigen can then go on to the second step, while the others die automatically. The B cells that internalize antigen are ranked

according to their binding energy, a proxy for the concentration of pMHC that they display to T cells, and only the best performers are selected.

*Equation 1* in the main text and the discussion that follows therein describes our model for the effects of CDR and FWR mutations on the binding free energy for antibody binding to variant antigens.

## Recycling to the dark zone, exit for differentiation, and termination of the GC reaction

As described in main text.

### Example: Affinity maturation against a single antigen

Our model depends on several parameters. Some of these are known a priori, e.g. SHM rate and fate of CDR mutations. Others represent biological quantities which have not yet been measured experimentally; for these we can make reasonable guesses. The rest do not have direct biological meaning, and exist on account of the simplified nature of the model. All parameters were adjusted manually until a good fit to experimental data in the single antigen case was obtained; see *Figure 4—figure supplement 1*. Consistent with the qualitative nature of the model, we did not attempt to choose an 'optimal' set of parameters, for example by precisely fitting the typical number of mutations or GC duration in the single antigen case. A strength of our model is that the qualitative results are robust to some variation in the parameters.

The dynamics of the GC population depends on the overall probabilities that B cells will survive somatic hypermutation and selection, successfully compete for T cell help, and be recycled back into the GC. The survival and selection probabilities depend implicitly on mutation rates, fractions of lethal versus silent and affinity-affecting mutations, etc. These overall probabilities are more important to the qualitative shape of the GC dynamics than the precise values of the underlying parameters. We write $p_{help}$ (help_cutoff in the simulation code) as the proportion of successful B cells which are selected (i.e. those that receive T cell help). $p_{recycle}$ denotes the proportion of B cells which are recycled back into the GC. The probability that a B cell successfully internalizes an antigen $i$ that it encounters depends is given by *Equation 4* above.

For convenience, all free energies entering into the AM simulations and those shown in the figures are made nondimensional using $k_BT$. The growth rate of the GC is proportional to the product $p_{surv} \times p_{int} \times p_{help} \times p_{recycle}$, where $p_{surv}$ is the probability that a B cell survives somatic hypermutation without lethal or strongly deleterious mutations.

We roughly tuned these parameters (within known experimental constraints) so that the population experiences a sharp decline at the beginning of AM until a few beneficial mutations appear and allow survival of a few B cells. Then the population plateaus for about 20 days until enough good mutations accumulate to increase the binding energy dramatically. The population then rises quickly until it reaches 1,536 cells by around 60 days. Experimental studies of affinity maturation in the presence of a single antigen showed that the antibodies produced accumulate about 10 mutations and their binding affinity – which is exponentially proportional to the binding energy ($A = \exp(-E/k_BT)$) – increases 1,000-fold (*Tas et al., 2016*). With our chosen parameters, our model reproduces these features well (*Figure 4—figure supplement 1*).

### Breadth calculations

Every B cell divides twice per cycle and its daughter cells have only a small chance of generating a non-lethal mutation. Consequently, the GC contains sets of functionally identical B cells, called B cell clones. All cells of a clone have the same binding energies and rigidity. The size of a clone varies with time and depends on its properties as explained above. We use the clone with the largest number of cells at the end of the reaction as the representative to analyze the events that shaped the final characteristics of most of the antibodies generated by AM. In order to analyze the breadth, we compute the binding free energy of this largest clone against an artificial panel of 100 antigens different from those that the antibody matured against. Here we take the overlap parameter $\lambda$ and the binding strength with the conserved region to be the same as in the simulations, but the binding strength with the variable region is randomly selected for each new antigen in the panel. Here, the random binding strength is chosen from a log-normal distribution that is broader (standard deviation $\sigma = 1$) and shifted toward weaker binding (mean $\mu = 3$) compared to the one used to generate the

binding free energies for the founder B cell-Ab combinations. This choice is based on the assumption that variable regions for a broad cross-section of antigens should be more variable and less likely to bind strongly to the Ab than those encountered in the original host. As a proxy for breadth, we compute the median binding energy of each antibody over the maturation pathway with panel antigens (*Figure 4*).

## Robustness to variation in $\lambda$

The value of the overlap parameter we have used in the main text, $\lambda = 0.9$, is chosen to roughly represent a swarm of diversified but still highly similar antigens that B cells may encounter in a chronically infected individual. We tested the robustness of our model to changes in $\lambda$ by performing simulations identical to those described above, but with $\lambda = 0.8$, and with a smaller number of panel antigens (5) to prevent excessive frustration. We also tuned the antigen concentration in simulations to compare overall, qualitative patterns of Ab maturation at comparable survival rates. We find that the same qualitative maturation pathways are also observed in these simulations (*Figure 4—figure supplement 2*). However, due to the greater dissimilarity between antigens against which the Abs mature, there is stronger selection for flexibility (lower Q).

## Acknowledgements

Financial support for this work was provided by a grant from the Lawrence Livermore National Laboratory (AKC, MK, VO, JL) the Ragon Institute of MGH, MIT, and Harvard (AKC, JPB), and the CHARMM Development Project (MK, VO).

## Additional information

### Competing interests

Arup K Chakraborty: Senior editor, *eLife*. The other authors declare that no competing interests exist.

### Funding

| Funder | Grant reference number | Author |
| --- | --- | --- |
| Lawrence Livermore National Laboratory | LLC Award #B620960 | Victor Ovchinnikov<br>Joy E Louveau<br>Martin Karplus<br>Arup K Chakraborty |
| Ragon Institute of MGH, MIT and Harvard | MGH Internal Fund #214931 | Joy E Louveau<br>John P Barton<br>Arup K Chakraborty |
| CHARMM Development Project | | Victor Ovchinnikov<br>Martin Karplus |

The funders had no role in study design, data collection and interpretation, or the decision to submit the work for publication.

### Author contributions

Victor Ovchinnikov, Joy E Louveau, John P Barton, Conceptualization, Investigation, Methodology, Writing—original draft, Writing—review and editing; Martin Karplus, Arup K Chakraborty, Conceptualization, Methodology, Writing—original draft, Writing—review and editing

### Author ORCIDs

Victor Ovchinnikov (iD) http://orcid.org/0000-0002-1793-2352
John P Barton (iD) http://orcid.org/0000-0003-1467-421X
Arup K Chakraborty (iD) http://orcid.org/0000-0003-1268-9602

Decision letter and Author response
Decision letter https://doi.org/10.7554/eLife.33038.021
Author response https://doi.org/10.7554/eLife.33038.022

## Additional files

### Supplementary files

• Supplementary file 1. List of all parameters used in affinity maturation simulations. Values are given if they are constant in *all* simulations presented here. Values of parameters that were changed for different simulations are given in *Supplementary file 2*.
DOI: https://doi.org/10.7554/eLife.33038.016

• Supplementary file 2. Table of simulation parameters whose values vary between different simulations. Note that the energy values in the simulation code have the *opposite sign* of the physical energy values, as described in the main text. This is simply a sign convention in the code, and has no bearing on the dynamics of the simulation.
DOI: https://doi.org/10.7554/eLife.33038.017

• Transparent reporting form
DOI: https://doi.org/10.7554/eLife.33038.018

• Reporting standard 1
DOI: https://doi.org/10.7554/eLife.33038.019

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
