## [Decision Letter]

Thank you for submitting your article "Role of framework mutations and antibody flexibility in the evolution of broadly neutralizing antibodies" for consideration by *eLife*. Your article has been reviewed by three peer reviewers, and the evaluation has been overseen by a Reviewing Editor and Michel Nussenzweig as the Senior Editor. The following individual involved in review of your submission has agreed to reveal his identity: Siduo Jiang (Reviewer #2).

The reviewers have discussed the reviews with one another and the Reviewing Editor has drafted this decision to help you prepare a revised submission.

The reviewers and the editor all found the work very interesting, the topic pertinent and well presented. The experiments are well-designed and demonstrate very thorough coverage and understanding of affinity maturation, especially in the context of anti-HIV BnAbs. They point out a number of points that all need to be addressed.

Essential revisions:

While the explicit-solvent simulations are clearly described, the description of the maturation model needs to be seriously improved. As written, the theoretical basis and definition of the model, as well as computational protocols are presented in a very disorganized fashion. Given that the *eLife* readership is largely experimental, it is unlikely that the description will be accessible outside of a narrow group of computational researchers. For example, the main text states that the model is coarse-grained, yet the level of graining does not appear to be defined. Based on the Supplementary Information, it appears that this is a population model that has no spatial resolution. As another example, the description of the model starts with a definition of how the system was seeded. However, this is premature, since the model has not been defined. It appears that the actual computational protocol is not mentioned until late in the Supplementary Information (Subsection “Selection in the light zone”). With these types of major issues in the organization of the maturation model description, it is very difficult to determine the significance of the presented analysis. In summary, a concise general form description of the theoretical model should be provided, first. Next, the authors should discuss their specific implementation and parameterization of the model, prior to describing results. Throughout this description, the limits of the model should also be presented.

Subsection “Results of the MD simulations” the fourth paragraph is confusing and not technically correct as written. What you observe is that A27 in the 109L LC, which mutates in Ser in mature LC, does not hbond to G68, which results in an instability of the 3-residue insertion at 66abc. However, mutations of these 3 residues from Pro-Asp-Ile in 109L to Asp-Ser-Pro in mature rescues this instability. The claim is that instability of 66abc "appears to be a consequence of mutations in the CDR1…" (Subsection “Results of the MD simulations”), but to prove this, one would need to show that S27 in conjunction with 66(a)P-66(b)D-66(c) of 109L LC results in a stable conformation. I suggest a re-wording such as the following:

“In the 109L LC, residue 27 is an alanine, and the stabilizing hydrogen bonds do not form, which allows the FWR3 region to unfold locally near the three-residue insertion 66(a)P-66(b)D-66(c). In the mature PGT121 light chain, this insertion is mutated to 66(a)D-66(b)S-66(c)P (Figure 2—figure supplement 1), and in conjunction with A27S, allows the formation a stable conformation. However, it is possible that this conformation is only weakly stable, such that minor changes in the adjacent structure (such as the S27A mutation in the CDR1) could cause unfolding. Therefore, stability of FWR3 in the intermediate 109L LC appears to partially depend on the sequence identify of CDR1, which might seem surprising because the framework regions are thought to provide a relatively rigid scaffold to which variable and flexible CDR regions are attached. Other sequence differences…” (continue with remainder of paragraph).

Minor note: sentence beginning ("One would therefore expect…") is somewhat misplaced since all of the mutations discussed in the FWR are stabilizing and not destabilizing.

I am less convinced by the AM model starting subsection” A model of AM suggests that the binding affinity of the germline BCR to the conserved regions of the epitope determines the role of framework mutations on bnAb evolution”. First is the general concern that multiple parameter sets (even though for the chosen set, most are justified by publications albeit with assumptions and simplifications) could have produced the results in Figure 4—figure supplement 1. If #mutations and binding affinity is treated as the "training" data, a cross-validation with other experimentally measured values would have been preferred. The parameters are especially crucial since beyond a certain point, small steps as you are performing using an evolutionary algorithm cannot escape a deep local minimum, and the conclusions merely falls out of the math rather than representing real biology. I also have a specific concern with the energy function, which is that E0 is kept constant for each simulation, whereas the value should really reflect the flexibility of the epitope itself. For a cocktail of antigens, this value should be sampled from an empirical distribution. Equ. 1a also currently models changes in entropy of binding as a function of changes in the antibody alone, which is perfectly accurate since we are not considering mutations in the antigen, but this analysis can be extended.

I suggest an easy follow-up experiment that would serve to further validate the model and its chosen parameters, solidify the conclusion that entropy is central to the evolution, and provide insight into the importance of the epitope during AM. Namely, you nicely showed that the profile for a weak initial Ec is bimodal, and that the distribution changes drastically with a small change in λ (in the sensible direction equivalent to decreasing initial Ec, since the energy term is λ*Ec). If the model is accurate, you would expect that a high constant value of E0 to drastically shift the distribution toward PGT121-like antibodies, and that a low value of E0 toward 3BN60-like antibodies. Using the current value in Figure 4 as control, please try two simulations, doubling and halving E0, or altering E0 in those directions as you see appropriate. This is different than altering values of Ec or λ, since that energy product is a variable in the simulation.

Subsection “Example: Affinity maturation against a single antigen”: Please state explicitly whether the initial and constant parameters (or distributions thereof) in the energy function and evolutionary scheme for the 10 antigen cocktail is the same as those used for the single antigen in Figure 4—figure supplement 1. If not, please comment on using the values obtained from Figure 4—figure supplement 1 (or similar values for which you have data) when extended to 10 antigens.

---

## [Author Response]

The reviewers and the editor all found the work very interesting, the topic pertinent and well presented. The experiments are well-designed and demonstrate very thorough coverage and understanding of affinity maturation, especially in the context of anti-HIV BnAbs. They point out a number of points that all need to be addressed.

We are very pleased by this reaction to our work and have worked hard to address the specific points made by the reviewers. We believe that, with one exception, we have addressed all the points made by the reviewers. Our response to reviewer comments has improved the manuscript, and we thank the editor and the reviewers for their help. In view of this, we hope that the paper will now be considered acceptable for publication in *eLife*.

Essential revisions:

While the explicit-solvent simulations are clearly described, the description of the maturation model needs to be seriously improved. As written, the theoretical basis and definition of the model, as well as computational protocols are presented in a very disorganized fashion. Given that the eLife readership is largely experimental, it is unlikely that the description will be accessible outside of a narrow group of computational researchers. For example, the main text states that the model is coarse-grained, yet the level of graining does not appear to be defined. Based on the Supplementary Information, it appears that this is a population model that has no spatial resolution. As another example, the description of the model starts with a definition of how the system was seeded. However, this is premature, since the model has not been defined. It appears that the actual computational protocol is not mentioned until late in the Supplementary Information (Subsection “Selection in the light zone”). With these types of major issues in the organization of the maturation model description, it is very difficult to determine the significance of the presented analysis. In summary, a concise general form description of the theoretical model should be provided, first. Next, the authors should discuss their specific implementation and parameterization of the model, prior to describing results. Throughout this description, the limits of the model should also be presented.

Following the reviewers’ suggestion, in the main text, we have completely rewritten and reorganized the description of the model that we use to simulate the affinity maturation process. Each step of the biology is now noted, followed by a description of how this is represented in the simplified model and the approximations that have been made. The pertinent section of the Methods (in supplement) only contains some elaboration of technical details. We thank the reviewers for this suggestion, and hope that they now find this presentation to be clear.

Subsection “Results of the MD simulations” the fourth paragraph is confusing and not technically correct as written. What you observe is that A27 in the 109L LC, which mutates in Ser in mature LC, does not hbond to G68, which results in an instability of the 3-residue insertion at 66abc. However, mutations of these 3 residues from Pro-Asp-Ile in 109L to Asp-Ser-Pro in mature rescues this instability. The claim is that instability of 66abc "appears to be a consequence of mutations in the CDR1…" (Subsection “Results of the MD simulations”), but to prove this, one would need to show that S27 in conjunction with 66(a)P-66(b)D-66(c) of 109L LC results in a stable conformation. I suggest a re-wording such as the following:“In the 109L LC, residue 27 is an alanine, and the stabilizing hydrogen bonds do not form, which allows the FWR3 region to unfold locally near the three-residue insertion 66(a)P-66(b)D-66(c). In the mature PGT121 light chain, this insertion is mutated to 66(a)D-66(b)S-66(c)P (Figure 2—figure supplement 1), and in conjunction with A27S, allows the formation a stable conformation. However, it is possible that this conformation is only weakly stable, such that minor changes in the adjacent structure (such as the S27A mutation in the CDR1) could cause unfolding. Therefore, stability of FWR3 in the intermediate 109L LC appears to partially depend on the sequence identify of CDR1, which might seem surprising because the framework regions are thought to provide a relatively rigid scaffold to which variable and flexible CDR regions are attached. Other sequence differences…” (continue with remainder of paragraph).

We agree with the reviewers and have revised the paragraph as suggested.

Minor note: sentence beginning ("One would therefore expect…") is somewhat misplaced since all of the mutations discussed in the FWR are stabilizing and not destabilizing.

We have removed the sentence in question, in part because its essence is already implied in the revised paragraph from the previous comment.

I am less convinced by the AM model starting subsection” A model of AM suggests that the binding affinity of the germline BCR to the conserved regions of the epitope determines the role of framework mutations on bnAb evolution”. First is the general concern that multiple parameter sets (even though for the chosen set, most are justified by publications albeit with assumptions and simplifications) could have produced the results in Figure 4—figure supplement 1. If #mutations and binding affinity is treated as the "training" data, a cross-validation with other experimentally measured values would have been preferred.

We have used known parameters whenever possible as the reviewer notes above. The parameters that are used are now further detailed in tables in the supplement. As we have noted, the purpose of our model is not to quantitatively reproduce experimental or simulation results (which are also not quantitatively absolutely accurate); rather, our goal is to provide mechanistic insights into the qualitative behavior revealed by the MD simulations and available experimental results on the role of framework mutations, and make qualitative predictions that can be further tested by experiments. We do not assert that the values of the unknown parameters are the only ones that would reproduce the changes in binding affinity observed in experiments with mouse models. We are, however, encouraged that the same parameters that recapitulate this feature (training data, if you will) also capture the behavior pertinent to the number of acquired mutations (cross validation, if you will). A strength of our model is that the qualitative mechanistic information revealed by the model with regard to the role of framework mutations for bnAb evolution are robust to variations of the parameters (e.g., see below our new calculations recommended by the reviewer on varying the value of E_0_). Therefore, we feel encouraged to make predictions about the relative rates at which CDR and FWR mutations may be acquired for different bnAb lineages – hints of which are already seen in existing phylogenetic data. Finally, in our past work using simplified models of affinity maturation to study bnAb evolution (e.g., Wang et al., 2015), we found that if unknown parameters were chosen to reflect the qualitative trends for affinity maturation induced by a single antigen, the mechanistic insights and predictions emerging from the model for bnAb evolution were positively tested by new experiments. We have not added this discussion to the paper at this point, but if the reviewer so recommends, we could add it to the Discussion section of the paper. We await editorial guidance in this regard.

The parameters are especially crucial since beyond a certain point, small steps as you are performing using an evolutionary algorithm cannot escape a deep local minimum, and the conclusions merely falls out of the math rather than representing real biology.

We apologize for not being sure that we understand this point made by the reviewer. Perhaps, the points noted above and the following point address this remark. Our results clearly show that our evolutionary simulations are not trapped in any local minimum. In all circumstances that we have simulated, there are large changes in the flexibility, affinity to conserved residues of the epitope, or both, as affinity maturation ensues. These changes are also physically/biophysically consistent with expectations as you vary values of parameters (such as studies varying E_0_ recommended by the reviewer – see below).

I also have a specific concern with the energy function, which is that E0 is kept constant for each simulation, whereas the value should really reflect the flexibility of the epitope itself. For a cocktail of antigens, this value should be sampled from an empirical distribution. Equ. 1a also currently models changes in entropy of binding as a function of changes in the antibody alone, which is perfectly accurate since we are not considering mutations in the antigen, but this analysis can be extended.

The reviewer is correct that our model could be extended by carrying out simulations with a distribution of values of E_0_. We have not carried out such calculations because of two reasons: (1) We have no idea how to guess the form of the distribution that should be used. (2) We are unsure as to what issues pertinent to the main points of our paper such calculations would address? We would be happy to do calculations with a distribution of values of E_0_, but we would be grateful for specific editorial guidance on these two questions. We also note that our guess is that the distribution is likely to be narrow since the variant antigens share the conserved residues of the epitope, and any distribution of epitope flexibility would emerge from difference in the amino acids at the variable sites in the same epitope. This is the only comment made by the reviewer that we have not addressed, rather we have carried out related calculations varying E_0_ as per another excellent suggestion made by the reviewer (see below).

I suggest an easy follow-up experiment that would serve to further validate the model and its chosen parameters, solidify the conclusion that entropy is central to the evolution, and provide insight into the importance of the epitope during AM. Namely, you nicely showed that the profile for a weak initial Ec is bimodal, and that the distribution changes drastically with a small change in λ (in the sensible direction equivalent to decreasing initial Ec, since the energy term is λ*Ec). If the model is accurate, you would expect that a high constant value of E0 to drastically shift the distribution toward PGT121-like antibodies, and that a low value of E0 toward 3BN60-like antibodies. Using the current value in Figure 4 as control, please try two simulations, doubling and halving E0, or altering E0 in those directions as you see appropriate. This is different than altering values of Ec or λ, since that energy product is a variable in the simulation.

We thank the reviewer for this excellent suggestion. We have carried out precisely the calculations that were recommended. We have added a description of these calculations in the main text (appended below) and added two supplementary figures that show these results.

“The results reported in the text were carried out with a particular value of *E_0_*, which is the binding energy to diverse antigens that becomes accessible to flexible BCRs/antibodies. We have explored the effects of varying the value of this parameter on our results when the germline B cell does not bind strongly to the conserved residues of the epitope (weak initial *E_c_*). As Figure 4—figure supplement 3 shows, shifting *E_0_* to very strong or weak binding energies affects the preferred antibody maturation pathway. A significantly more favorable value of *E_0_* than that used to obtain the results in Figure 4 makes flexibility-increasing framework mutations more beneficial, and thus drives antibody evolution more along 3BNC60-like trajectories (compare Figure 4—figure supplement 3 with Figure 4). In this circumstance, antibodies only begin to rigidify and become more like the PGT121 lineage later in the maturation cycle, after mutations that make *E_c_*(binding energy to the conserved residues) very strong have been acquired. So, if *E_0_* is stronger than what was used to obtain the results in Figure 4, the qualitative behavior does not change, but the 3BNC60like antibodies are more likely to be observed for a greater duration of affinity maturation and are likely to be more potent. If *E_0_* is weaker than what was used to obtain the results in Figure 4, and we use the same antigen concentration as before, most of the B cells die and the GC is likely to be extinguished. This is because affinity maturation is frustrated when there are multiple variant antigens and the initial *E_c_* is weak, as has been described before (Wang et al., 2015; Shaffer et al., 2016), and a path to relieving this frustration by evolving flexibility through FWR mutations is no longer available (because of weak E_0_). In this circumstance, if the antigen concentration is increased, we find that the GCs are not extinguished with high probability, and more CH103-like maturation trajectories are promoted (compare Figure 4—figure supplement 3 with Figure 4). This is because CDR mutations that improve binding with the conserved region (stronger *E_c_*) become relatively more beneficial early in the maturation process since *E_0_* is weak and there is little benefit to evolving FWR mutations that affect flexibility. Consistent with the points noted above, varying values of *E_0_* also change the rate of acquiring CDR versus FWR mutations (Figure 6—figure supplement 1), with stronger values of *E_0_* promoting FWR mutations more than in Figure 6 and weaker values of *E_0_* suppressing FWR mutations.”

Subsection “Example: Affinity maturation against a single antigen”: Please state explicitly whether the initial and constant parameters (or distributions thereof) in the energy function and evolutionary scheme for the 10 antigen cocktail is the same as those used for the single antigen in Figure 4—figure supplement 1. If not, please comment on using the values obtained from Figure 4—figure supplement 1 (or similar values for which you have data) when extended to 10 antigens.

Precisely the same parameters were used. However, the antigen concentration was increases to study the cocktail. This is because, at the same antigen concentrations, as has been described in this paper and previously, for a cocktail of variant antigens the frustration level becomes too high leading to extinction of many GCs. Vaccination with a cocktail would require higher antigen concentrations than what is optimal for a single antigen. This point is now addressed in the new tables of parameters.